# Transformers as Stochastic Optimizers

**Ryuichiro Hataya** [1]  **Masaaki Imaizumi** [2] [1]

## Abstract

In-context learning is a crucial framework for understanding the learning processes of foundation models. Transformers are frequently used as a useful architecture within this context. Recent experimental results have demonstrated that Transformers can learn algorithms such as gradient descent based on datasets. However, from a theoretical aspect, while Transformers have been shown to approximate non-stochastic algorithms, it has not been shown for stochastic algorithms such as stochastic gradient descent. This study develops a theory on how Transformers represent stochastic algorithms in in-context learning. Specifically, we show that Transformers can generate truly random numbers by extracting the randomness inherent in the data and pseudo-random numbers by implementing pseudo-random number generators. As a direct application, we demonstrate that Transformers can implement stochastic optimizers, including stochastic gradient descent and Adam, in context.

## 1. Introduction

Among various strong capabilities of foundation models, their in-context learning ability is powerful and thus actively investigated. Using in-context learning, foundation models, typically large language models, can perform new tasks presented at test time without updating their parameters. Such a learning ability is not only observed empirically (Garg et al., 2022; Von Oswald et al., 2023; Akyürek et al., 2022) but also analyzed theoretically (Li et al., 2023; Xie et al., 2021; Zhang et al., 2023; Bai et al., 2023; Lin et al., 2023; Ahn et al., 2024; Raventós et al., 2024), revealing that Transformers can approximate learning algorithms, such as least square (Zhang et al., 2023) and gradient descent (Akyürek et al., 2022). In particular, (Bai et al., 2023) showed that a

---
[1]RIKEN AIP, Tokyo, Japan [2]The University of Tokyo, Tokyo, Japan. Correspondence to: Ryuichiro Hataya <ryuichiro.hataya@riken.jp>.

*Proceedings of the 1st Workshop on In-Context Learning at the 41st International Conference on Machine Learning*, Vienna, Austria. 2024. Copyright 2024 by the author(s).

Transformer layer can approximate a single step of gradient descent of linear models, and thus, Transformers can perform training of linear models. Although these results are powerful, the approximable algorithms are *non-stochastic*.

This paper unveils that Transformers can indeed approximate *stochastic* algorithms by generating random numbers in context. Specifically, we show that Transformers can construct random numbers 1. extracting randomness in randomly sampled input data; and 2. implementing pseudo-random number generators, such as Mersenne Twister. As a direct application, we extend the in-context gradient descent to in-context *stochastic* gradient descent. We further show that Transformers can represent more complex optimizers, such as Adam, which further empowers ICL.

## 2. Preliminary

### Notation

$\mathbf{0}_A, \mathbf{1}_A$ indicate $A$-dimensional vectors all of whose elements are $0$ or $1$. The notations for elementwise operators for vectors are often abused for brevity, *e.g.*, for a vector $\boldsymbol{a}$, $1/\boldsymbol{a}$, $\boldsymbol{a}^a$, $\sqrt{\boldsymbol{a}}$ denote elementwise division, power by a, and square root, respectively. To measure the distance between the distributions of $\boldsymbol{x}, \boldsymbol{x}' \in \mathbb{R}^p$, we use Kormogorov distance $\Delta(\boldsymbol{x}, \boldsymbol{x}') = \sum_{A \subset \mathbb{R}^p} |\Pr(\boldsymbol{x} \in A) - \Pr(\boldsymbol{x}' \in A)|$, where $A$ is taken from all measurable set in the parameter space $\mathbb{R}^p$.

### 2.1. Transformer

Define an $L$-layer Transformer consisting of $L$ Transformer layers as follows. The $l$th Transformer layer maps an input matrix $\boldsymbol{H}^{(l)} \in \mathbb{R}^{D \times N}$ to $\hat{\boldsymbol{H}}^{(l)} \in \mathbb{R}^{D \times N}$ and is composed of a self-attention block and a feed-forward block. The self-attention block $\mathrm{Attn}^{(l)} : \mathbb{R}^{D \times N} \to \mathbb{R}^{D \times N}$ is parameterized by $D \times D$ matrices $\{(\boldsymbol{K}_m^{(l)}, \boldsymbol{Q}_m^{(l)}, \boldsymbol{V}_m^{(l)})\}_{m=1}^M$, where $M$ is the number of heads, and defined as

$$\mathrm{Attn}^{(l)}(\boldsymbol{X}) = \boldsymbol{X} + \frac{1}{N} \sum_{m=1}^M \boldsymbol{V}_m^{(l)} \boldsymbol{X} \sigma((\boldsymbol{Q}_m^{(l)} \boldsymbol{X})^\top \boldsymbol{K}_m^{(l)} \boldsymbol{X}). \tag{1}$$

$\sigma$ denotes an activation function applied elementwisely.

The feed-forward block $\mathrm{MLP}^{(l)} : \mathbb{R}^{D \times N} \to \mathbb{R}^{D \times N}$ is a multi-layer perceptron with a skip connection, parameter-

ized by $(\boldsymbol{W}_1^{(l)}, \boldsymbol{W}_2^{(l)}) \in \mathbb{R}^{D' \times D} \times \mathbb{R}^{D \times D'}$, such that

$$\mathrm{MLP}^{(l)}(\boldsymbol{X}) = \boldsymbol{X} + \boldsymbol{W}_2^{(l)}\varsigma(\boldsymbol{W}_1^{(l)}\boldsymbol{X}), \qquad (2)$$

where $\varsigma$ is an activation function applied elementwisely. We let both $\sigma$ and $\varsigma$ the ReLU function in this paper.

In summary, an $L$-layer Transformer $\mathrm{TF}_{\boldsymbol{\theta}}$, parameterized by $\boldsymbol{\theta}_m^{(l)} := (\boldsymbol{K}_m^{(l)}, \boldsymbol{Q}_m^{(l)}, \boldsymbol{V}_m^{(l)})$ and

$$\boldsymbol{\theta} = \{(\boldsymbol{\theta}_1^{(l)}, \dots, \boldsymbol{\theta}_M^{(l)}, \boldsymbol{W}_1^{(l)}, \boldsymbol{W}_2^{(l)})\}_{l=1}^L, \qquad (3)$$

is a composition of the abovementioned layers as

$$\mathrm{TF}_{\boldsymbol{\theta}}(\boldsymbol{X}) = \mathrm{MLP}^{(L)} \circ \mathrm{Attn}^{(L)} \circ \cdots \circ \mathrm{MLP}^{(1)} \circ \mathrm{Attn}^{(1)}(\boldsymbol{X}). \qquad (4)$$

In the remaining text, the superscript to indicate the number of layer $^{(l)}$ is sometimes omitted for brevity. In some cases, we denote the $n$th columns of $\boldsymbol{H}^{(l)}, \tilde{\boldsymbol{H}}^{(l)}$ as $\boldsymbol{h}_n^{(l)}, \tilde{\boldsymbol{h}}_n^{(l)}$. We define the following norm of a Transformer $\mathrm{TF}_{\boldsymbol{\theta}}$:

$$\|\boldsymbol{\theta}\|_{\mathrm{TF}} = \max_{l \in \{1,\dots,L\}} \left\{ \max_{m \in \{1,\dots,M\}} \left\{ \|\boldsymbol{Q}_m^{(l)}\|, \|\boldsymbol{K}_m^{(l)}\| \right\} \quad (5) \right.$$

$$\left. + \sum_{m=1}^M \|\boldsymbol{V}_m^{(l)}\| + \|\boldsymbol{W}_1^{(l)}\| + \|\boldsymbol{W}_2^{(l)}\| \right\}, \qquad (6)$$

where $\|\cdot\|$ for matrices indicates the operator norm in this equation.

## 2.2. In-context Learning

In the in-context learning (ICL), a virtual model is given a dataset $\mathcal{D} = \{(\boldsymbol{x}_i, y_i)\}_{i=1}^N \sim (P)^N$ and a test data point $\boldsymbol{x}_*$ from a marginal distribution $P_{\boldsymbol{x}}$ and then predicts its label $y_*$. The dataset consists of $N$ pairs of inputs $\boldsymbol{x}_i \in \mathbb{R}^d$ and its label $y_i \in \mathbb{R}$ Our goal is to construct a fixed Transformer to perform ICL, by learning an algorithm for the virtual model to predict $y_*$ using $(\mathcal{D}_j, \boldsymbol{x}_{*j})$ sampled from different distributions $P_j$ from $P$.

The input dataset and the test data point are encoded into $\boldsymbol{H}^{(1)} \in \mathbb{R}^{D \times (N+1)}$ as follows:

$$\boldsymbol{H}^{(1)} = \begin{bmatrix} \boldsymbol{x}_1 & \boldsymbol{x}_2 & \dots & \boldsymbol{x}_N & \boldsymbol{x}_* \\ y_1 & y_2 & \dots & y_N & 0 \\ 1 & 1 & \dots & 1 & 1 \\ t_1 & t_2 & \dots & t_N & t_{N+1} \\ \boldsymbol{0}_{D-(d+p+2)} & \boldsymbol{0}_{D-(d+p+2)} & \dots & \boldsymbol{0}_{D-(d+p+2)} & \boldsymbol{0}_{D-(d+p+2)} \\ \boldsymbol{p}_1 & \boldsymbol{p}_2 & \dots & \boldsymbol{p}_N & \boldsymbol{p}_{N+1} \end{bmatrix}, \qquad (7)$$

where $t_n = 1$ for $n \leq N$ and $t_{N+1} = 0$ is used to indicate which data points are from $\mathcal{D}$. $\boldsymbol{p}_n \in \mathbb{R}^p$ encodes the position information. Using these notations, the goal of ICL can be rewritten as predicting $y_*$ by $\tilde{\boldsymbol{H}}_{N+1,1}^{(L)}$, where $\tilde{\boldsymbol{H}}^{(L)} = \mathrm{TF}_{\boldsymbol{\theta}}(\boldsymbol{H}^{(1)})$, by the acquired algorithm.

## 2.3. In-context Gradient Descent

Bai et al. demonstrated that Transformers could implement *non-stochastic* gradient descent of a linear model for a broad class of convex loss functions in an in-context way. The key ingredient is the following approximability.

**Definition 1** (($\varepsilon, R, M, C$)-approximability by sum of Re-LUs, (Bai et al., 2023)). *For $\varepsilon > 0$ and $R \geq 1$, a function $g : \mathbb{R}^k \to \mathbb{R}$ is ($\epsilon, R, M, C$)-approximabile by sum of ReLUs if there exist a function $f(\boldsymbol{z}) = \sum_{m=1}^M c_m \sigma(\boldsymbol{a}_m^\top \boldsymbol{z} + b_m)$ with $\sum_{m=1}^M |c_m| \leq C$, $\max_{m \in \{1,\dots,M\}} \|\boldsymbol{a}\|_1 + b_m \leq 1$, where $\boldsymbol{a}_m \in \mathbb{R}^k, b_m \in \mathbb{R}, c_m \in \mathbb{R},$, such that $\sup_{\boldsymbol{z} \in [-R,R]^k} |g(\boldsymbol{z}) - f(\boldsymbol{z})| \leq \varepsilon$.*

This notion enables the attention block (1) and the MLP block (2) to approximate various functions, including loss functions:

**Theorem 1** (Theorem 9 of (Bai et al., 2023)). *Fix any $B_w > 0$, $L > 1$, $\eta > 0$, $K > 0$, and $\epsilon \leq B_w/2L$. Given a loss function $\ell$ that is convex in the first argument, and $\nabla_1 \ell$ is ($\epsilon, R, M, C$)-approximable by the sum of ReLUs with $R = \max(B_w, B_x, B_y, 1)$. Let $\boldsymbol{h}_n^{(1)} = [\boldsymbol{x}_n, y_n, 1, t_n, \boldsymbol{0}_{D-(d+p+3)}, \boldsymbol{p}_n]$ for $n = 1, 2, \dots, N+1$. Then, there exists an attention-only Transformer $\mathrm{TF}_{\boldsymbol{\theta}}$ with $(L+1)$ layers and $M$ heads such that for any input $(\mathcal{D}, \boldsymbol{x}_*)$ such that $\sup_{\boldsymbol{w}:\|\boldsymbol{w}\|_2 \leq B_w} \lambda_{\max}(\nabla^2 \hat{L}(\boldsymbol{w}; \mathcal{D})) \leq 2/\eta$ and $\exists \boldsymbol{w}^\star \in \arg\min_{\boldsymbol{w} \in \mathbb{R}^d} \hat{L}(\boldsymbol{w}; \mathcal{D})$ such that $\|\boldsymbol{w}^\star\|_2 \leq B_w/2$, $\mathrm{TF}_{\boldsymbol{\theta}}$ approximately implements IC-GD with initialization $\boldsymbol{w}_{\mathrm{GD}}^{(0)} = \boldsymbol{0}_d$: For every $l \in \{1, \dots, L\}$, the lth layer's output $\tilde{\boldsymbol{H}}^{(l)}$ approximates $l$ steps of IC-GD: we have $\boldsymbol{h}_n^{(l)} = [\boldsymbol{x}_n, y_n, 1, t_n, \hat{\boldsymbol{w}}^{(l)}, \boldsymbol{0}_{D-(L+2d+p+2)}, \boldsymbol{p}_1]$ for each $n \in \{1, \dots, N\}$, where $\|\hat{\boldsymbol{w}}^{(l)} - \boldsymbol{w}_{\mathrm{GD}}^{(l)}\|_2 \leq \epsilon l \eta B_x$. The Transformer also admits norm bound $\|\boldsymbol{\theta}\|_{\mathrm{TF}} \leq 2 + R + 2\eta C$.*

## 3. In-context Random Number Generation

In this section, we show that Transformers can generate random numbers in two ways.

### 3.1. Generating *Truly* Random Numbers

First, we demonstrate that a single layer Transformer can generate a truly random number on $[0, 1]$ using the stochasticity in data. The key idea is to estimate the density function of data using some training data points and then evaluate it with a held-out data point.

**Theorem 2** (Generating a Random Number). *For any $\epsilon > 0$ and $B_x > 0$, there exists a self-attention block $\mathrm{Attn}_{\boldsymbol{\theta}}$ with two heads and $\|\boldsymbol{\theta}\|_{\mathrm{TF}} \leq \frac{7}{2} + \max\{\frac{1}{4\epsilon} + 2, (B_x + 1)\frac{1}{4\epsilon}\}$ such that, for any input $(\mathcal{D}, \boldsymbol{x}_*)$, $\mathrm{TF}_{\boldsymbol{\theta}}$ approximately implements the cumulative distribution function $\hat{P}_z(z)$ of $\{z_1, \dots, z_{N-1}\}$, where $z_n = \boldsymbol{x}_{n,1} \sim P_{x,1}$, such that, for*

$$z_N = \boldsymbol{x}_{N,1},$$

$$\Delta(\hat{P}_z(z_N), u) \leq \epsilon + \mathcal{O}(\frac{1}{\sqrt{N}}), \qquad (8)$$

*for $u \sim \mathcal{U}(0,1)$.*

### 3.2. Generating *Pseudo* Random Numbers

Next, we show that Transformers can implement pseudo-random number generators, including Mersenne Twister (Matsumoto & Nishimura, 1998), which is a popular pseudo-random number generator: for example, Python's `random` module adopts it[1].

**Definition 2** (Pseudo Random Number Generator over $\mathbb{F}_2$). The following linear generator over the finite field of order 2, $\mathbb{F}_2$, outputs a pseudo-random number $\boldsymbol{o}_t \in \mathbb{F}_2^w$ given a state $\boldsymbol{s}_t \in \mathbb{F}_2^k$

$$\boldsymbol{s}_t = \boldsymbol{A}\boldsymbol{s}_{t-1},$$
$$\boldsymbol{o}_t = \boldsymbol{B}\boldsymbol{s}_t,$$

where $\boldsymbol{A} \in \mathbb{F}_2^{k \times k}$, and $\boldsymbol{B} \in \mathbb{F}_2^{k \times m}$, for $t \in \mathbb{N}$. $\boldsymbol{s}_0$ is the seed.

By selecting $\boldsymbol{A}$ and $\boldsymbol{B}$ appropriately, this generalized generator obtains several pseudo-random number generators, such as Mersenne Twister.

**Theorem 3** (Implementing Pseudo-random Number Generator). *For any state $\boldsymbol{s}_0 \in \mathbb{F}_2^k$, a single self-attention block with $M$ heads can generate $(\boldsymbol{o}_1, \ldots, \boldsymbol{o}_M)$ exactly using the pseudo-random number generators in Definition 2.*

We can generate pseudo-random numbers by using a random number generated in Theorem 2 as an initial seed. This is basically the same as what we do in numerical experiments.

## 4. In-context *Stochastic* Gradient Descent

In this section, we extend in-context gradient descent in Theorem 1 to in-context *stochastic* gradient descent. We assume that the following function can be constructed in context.

**Assumption 1.** *Fix a sequence of (pseudo-) random numbers $(u_1, \ldots, u_K)$. There exists a Transformer $\mathrm{TF}_{\boldsymbol{\theta}}$ such that maps input $\boldsymbol{h}_n^{(1)} = [\boldsymbol{x}_n, y_n, 1, t_n, u, \boldsymbol{0}_n, \boldsymbol{0}_{D-(d+p+3)}, \boldsymbol{p}_n]$ to $\boldsymbol{h}_n^{(1)} = [\boldsymbol{x}_n, y_n, 1, t_n, u, \boldsymbol{b}_n, \boldsymbol{0}_{D-(d+p+3)}, \boldsymbol{p}_n]$ for $n = 1, 2, \ldots, N$, where $\boldsymbol{b}_n \in \{0,1\}^L$ determines a minibatch of size $K$ such that $b_{n,l} = 1$ indicates that the $n$th data point is in the minibatch at the $l$th iteration.*

[1] https://docs.python.org/3/library/random.html

Here, the positional information $\boldsymbol{p}_n$ is used to assign $\boldsymbol{b}_n$ to each $n$.

For the approximation, we define a sequence of parameters $\{\boldsymbol{w}_{\mathrm{SGD}}^{(l)}\}_{l=1,\ldots,L}$ generated by stochastic gradient descent:

$$\boldsymbol{w}_{\mathrm{SGD}}^{(l+1)} = \boldsymbol{w}_{\mathrm{SGD}}^{(l)} - \frac{\eta}{K} \sum_{(\boldsymbol{x},y) \in \mathcal{B}_l} \nabla_{\boldsymbol{w}} \ell(\boldsymbol{w}_{\mathrm{SGD}}^{(l)\top} \boldsymbol{x}, y), \quad (9)$$

where $\eta$ is a learning rate, $\mathcal{B}_l$ is a minibatch of size $K$ for the $l$th iteration, and $\ell : \mathbb{R} \times \mathbb{R} \to \mathbb{R}^{\geq 0}$ is a loss function. The trained model is evaluated by $f(\boldsymbol{x}; \boldsymbol{w}) = \boldsymbol{w}^\top \boldsymbol{x}$. We suppose that $\boldsymbol{x}_n \leq B_w$, $y \leq B_y$, and $\boldsymbol{w}^{(l)} \leq B_w$, for each $n$ and $l$.

**Theorem 4** (Implementation of In-context *Stochastic* Gradient Descent). *Fix any $B_w > 0$, $L > 1$, $\eta > 0$, $K > 0$, and $\epsilon \leq B_w/2L$. Given a loss function $\ell$ that is convex in the first argument, and $\nabla_1 \ell$ is $(\epsilon, R, M, C)$-approximable by the sum of ReLUs with $R = \max(B_w, B_w, B_y, 1)$. Let $\boldsymbol{h}_n^{(1)} = [\boldsymbol{x}_n, y_n, 1, t_n, u, \boldsymbol{b}_n, \boldsymbol{0}_{D-(d+p+3)}, \boldsymbol{p}_n]$ for $n = 1, 2, \ldots, N$. Then, there exists a Transformer $\mathrm{TF}_{\boldsymbol{\theta}}$ with $(L+1)$ layers and $M$ heads such that for any input $(\mathcal{D}, \boldsymbol{x}_*)$ such that $\sup_{\boldsymbol{w}:\|\boldsymbol{w}\|_2 \leq B_w} \lambda_{\max}(\nabla^2 \hat{L}(\boldsymbol{w}; \mathcal{B})) \leq 2/\eta$ and $\exists \boldsymbol{w}^\star \in \operatorname{argmin}_{\boldsymbol{w} \in \mathbb{R}^d} \hat{L}(\boldsymbol{w}; \mathcal{B})$ such that $\|\boldsymbol{w}^\star\|_2 \leq B_w/2$ for any $\mathcal{B} \sim \mathcal{D}$ with a minibatch size of $K$, $\mathrm{TF}_{\boldsymbol{\theta}}$ approximately implements SGD with initialization $\boldsymbol{w}_{\mathrm{SGD}}^{(0)} = \boldsymbol{0}_d$:*

*For every $l \in \{1, \ldots, L\}$, the $l$th layer's output $\tilde{\boldsymbol{H}}^{(l)}$ approximates $l$ steps of SGD: we have $\boldsymbol{h}_n^{(l)} = [\boldsymbol{x}_n, y_n, t_n, 1, u, \boldsymbol{b}_n, \hat{\boldsymbol{w}}^{(l)}, \boldsymbol{0}_{D-(L+2d+p+2)}, \boldsymbol{p}_n]$ for each $n \in \{1, \ldots, N\}$, where*

$$\Delta(\hat{\boldsymbol{w}}^{(l)}, \boldsymbol{w}_{\mathrm{SGD}}^{(l)}) \leq \epsilon l \eta B_x. \qquad (10)$$

*As a result, it approximates the output for a test data point as*

$$\Delta(f(\boldsymbol{x}_*, \boldsymbol{w}_{\mathrm{SGD}}^{(L)}), \mathrm{TF}_{\boldsymbol{\theta}}(\boldsymbol{H}^{(1)})) \leq \epsilon L \eta B_x^2. \qquad (11)$$

*Such a Transformer admits $\|\boldsymbol{\theta}\|_{\mathrm{TF}} \leq 2 + R + 2\eta C$.*

Additionally, we present that Transformers can approximate some (adaptive) first-order stochastic optimizers, such as Adam (Kingma & Ba, 2015).

Let a sequence of parameters $\{\boldsymbol{w}_{\mathrm{Adam}}^{(l)}\}_{l=1,\ldots,L}$ generated by Adam as follows:

$$\boldsymbol{w}_{\mathrm{Adam}}^{(l+1)} = \boldsymbol{w}_{\mathrm{Adam}}^{(l)} - \eta \frac{\boldsymbol{m}^{(l)}/(1-\beta_1^l)}{\sqrt{\boldsymbol{v}^{(l)}/(1-\beta_2^l)} + \varepsilon \mathbf{1}}, \qquad (12)$$

$$\boldsymbol{m}^{(l)} = \beta_1 \boldsymbol{m}^{(l-1)} + (1-\beta_1)\boldsymbol{g}, \qquad (13)$$

$$\boldsymbol{v}^{(l)} = \beta_2 \boldsymbol{v}^{(l-1)} + (1-\beta_2)\boldsymbol{g}^2, \qquad (14)$$

$$\boldsymbol{g} = \frac{1}{K} \sum_{(\boldsymbol{x},y) \in \mathcal{B}_l} \nabla_{\boldsymbol{w}} \ell(\boldsymbol{w}_{\mathrm{Adam}}^{(l)\top} \boldsymbol{x}, y), \qquad (15)$$

where $\eta > 0$ is a learning rate, $\beta_1, \beta_2 \in [0, 1)$ are decay rates, $\varepsilon > 0$ is a small constant to avoid division by zero, and $\boldsymbol{m}^{(l)}, \boldsymbol{v}^{(l)} \in \mathbb{R}^d$ are buffers, initialized by zeros.

**Theorem 5** (Implementation of Adam)**.** *Fix any* $B_w > 0$, $L > 1$, $\eta > 0$, $K > 0$, *and* $\epsilon \leq B_w/2L$. *Given a loss function and* $\boldsymbol{h}_n^{(1)}$ *in Theorem 4. Then, there exists a Transformer* $\mathrm{TF}_{\boldsymbol{\theta}}$ *with* $2L+1$ *layers with* $M$ *heads self-attention blocks and feed-forward blocks with width* $D'$ *such that for any inputs* $(\mathcal{D}, \boldsymbol{x}_*)$ *in Theorem 4,* $\mathrm{TF}_{\boldsymbol{\theta}}$ *approximately implements IC-Adam with initialization* $\hat{\boldsymbol{w}}_{\mathrm{Adam}}^{(0)} = \boldsymbol{0}_d$: *For every* $l \in \{2, \ldots, L\}$, *the* $2l$*th layer's output* $\tilde{\boldsymbol{H}}^{(2l)}$ *approximates* $l$ *steps of IC-Adam: we have* $\boldsymbol{h}_n^{(2l)} = [\boldsymbol{x}_n, y_n, 1, u, \boldsymbol{b}_n, \hat{\boldsymbol{w}}^{(l)}, \beta_1 \hat{\boldsymbol{m}}^{(l)}, \beta_2 \hat{\boldsymbol{v}}^{(l)}, \boldsymbol{0}_{D-(L+4d+p+2)}, \boldsymbol{p}_n]$ *for every* $n \in \{1, \ldots, N\}$, *where*

$$\Delta(\hat{\boldsymbol{w}}^{(l)}, \boldsymbol{w}_{\mathrm{Adam}}^{(l)}) \leq \epsilon l \eta B_x. \tag{16}$$

*The norm of the Transformer admits* $\|\boldsymbol{\theta}\|_{\mathrm{TF}} \leq \max\{5 + R + 2C + \beta_2 + \frac{2}{M_2} + (1-\beta_2)C_2, \frac{1}{1-\max(\beta_1,\beta_2)} + \eta C_3\}$.

**Remark 1.** By using Theorem 5, we can show that Transformers can implement other optimizers, such as Momentum SGD, Adagrad, and RMSProp.

## 5. Proof Outline

**Proof outline of Theorem 2** We can construct the cumulative distribution function $\hat{P}_z(t) = \frac{1}{N-1} \sum_{n=1}^{N-1} \mathbb{1}_{z \leq t}$. This function can be approximated by the sum of ReLUs as

$$\hat{P}_z(t) = \frac{1}{N-1} \sum_{n=1}^{N-1} \{\sigma(a(z_n - t) + 0.5) + \sigma(a(z - t) - 0.5)\}, \tag{17}$$

where $a = \frac{1}{4\epsilon} > 0$. This function can be represented by a self-attention block.

**Proof outline of Theorem 3** The $m$th head of the self-attention block can contain $\boldsymbol{B}\boldsymbol{A}^m$ for $m = 1, \ldots, M$, outputting $\boldsymbol{o}_m = \boldsymbol{B}\boldsymbol{A}^m \boldsymbol{s}_0$.

**Proof outline of Theorem 4** We use the $(\epsilon, R, M, C)$-approximability of $(s, t) \mapsto \partial_1 \ell(s, t)$ at the $l$th iteration by the sum of ReLUs to approximate $\partial_1 \ell(\boldsymbol{w}^\top \boldsymbol{x}, y)$ as $f(\boldsymbol{w}^\top \boldsymbol{x}, y) = \sum_{m=1}^{M} c_m \sigma(a_m \boldsymbol{w}^\top \boldsymbol{x} + b_m y + d_m - R(1 - b_{n,l}))$, where $R = \max(B_x B_w, B_y, 1)$, so that $f(\boldsymbol{w}^\top \boldsymbol{x}, y) = 0$ if $b_{n,l} = 0$.

**Proof outline of Theorem 5** We use the $(\epsilon, R, M, C)$-approximability of $(s, t) \mapsto \partial_1 \ell(s, t)$, $s \mapsto s^2$, and $(s, t) \mapsto \frac{s/(1-\beta_1^l)}{\sqrt{t/(1-\beta_2^l)+\varepsilon}}$.

## 6. Conclusion and Discussion

In this work, we have demonstrated the capabilities of the in-context learning framework to implement random number generation and stochastic gradient descent algorithms.

Our findings broaden the applications of in-context learning, extending its reach to stochastic algorithms, which possess unique advantages over their non-stochastic counterparts. Notably, stochastic algorithms can solve certain problems that non-stochastic algorithms cannot address effectively. For instance, stochastic gradient descent has an asymptotic global convergence guarantee for sufficiently regular non-convex objectives (Raginsky et al., 2017), a property that non-stochastic gradient descent methods lack. While our work showcases the potential of in-context learning for stochastic algorithms, exploring its application to more complex scenarios remains an intriguing avenue for future research.

Theorem 2 constructs an empirical distribution function using $N - 1$ training data points and generates a random number with another data point. As a result, if the order of training data changes, the generated random number also changes. This aligns with the empirical observation that the order of prompts alters the performance (Lu et al., 2022). Further investigation of this line is also an interesting direction.

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

*Proof of Theorem 2.* The empirical cumulative distribution function $P_z(t)$ can be defined as $P_z(t) = \frac{1}{N}\sum_{n=1}^{N} \mathbb{1}_{\boldsymbol{x}_{n,0} \leq t}$. This function can be approximated by sum of ReLU functions as

$$\hat{P}_z(t) = \frac{1}{N}\sum_{n=1}^{N}\{\sigma(a(\boldsymbol{x}_{n,1} - t) + 0.5) + \sigma(a(\boldsymbol{x}_{n,1} - t) - 0.5)\}, \tag{18}$$

where $a = \frac{1}{4\epsilon} > 0$. Equation (18) can be represented by self-attention block with matrices $\boldsymbol{Q}_m, \boldsymbol{K}_m, \boldsymbol{V}_m$ for $m = \pm$, such that

$$\boldsymbol{Q}_m \boldsymbol{h}_i = \begin{bmatrix} a\boldsymbol{x}_{i,0} \pm 0.5 \\ 1 \\ -2 \\ \boldsymbol{0}_{D-3} \end{bmatrix}, \quad \boldsymbol{K}_m \boldsymbol{h}_j = \begin{bmatrix} 1 \\ -a\boldsymbol{x}_{j,0} \\ (aB_x \pm 0.5)t_j \\ \boldsymbol{0}_{D-3} \end{bmatrix}, \quad \text{and } \boldsymbol{V}_m \boldsymbol{h}_j = \begin{bmatrix} \boldsymbol{0}_{d+3} \\ (N+1)/N \\ \boldsymbol{0}_{D-(d+4)} \end{bmatrix}, \tag{19}$$

For $\boldsymbol{h}_i = [\boldsymbol{x}_i, y_i, 1, t_i, \boldsymbol{0}, \boldsymbol{p}_i]$, such matrices exist and can be bounded as $\max_m \|\boldsymbol{Q}_m\| \leq a + \frac{7}{2}$, and $\max_m \|\boldsymbol{K}_m\| \leq (B_x + 1)a + \frac{3}{2}$, $\sum_m \|\boldsymbol{V}_m\| \leq 2$, and thus $\|\boldsymbol{\theta}\|_{\text{TF}} \leq \frac{7}{2} + \max\{\frac{1}{4\epsilon} + 2, (B_x + 1)\frac{1}{4\epsilon}\}$. Then,

$$\sigma(\langle \boldsymbol{Q}_m \boldsymbol{h}_i, \boldsymbol{K}_m \boldsymbol{h}_j \rangle) \tag{20}$$

$$= \sigma(a(\boldsymbol{x}_{i,1} - \boldsymbol{x}_{j,1} \pm 0.5) - (aB_x \pm 0.5)t_j) \tag{21}$$

$$= \begin{cases} 0 & \text{if } j \leq N \\ \sigma(a(\boldsymbol{x}_{i,1} - \boldsymbol{x}_{*,1}) \pm 0.5) \end{cases}. \tag{22}$$

Consequently, we get

$$\sum_{i=1}^{N+1}\sum_{m=\pm} \sigma(\langle \boldsymbol{Q}_m \boldsymbol{h}_i, \boldsymbol{K}_m \boldsymbol{h}_j \rangle)\boldsymbol{V}_m \boldsymbol{h}_j \tag{23}$$

$$= \frac{N+1}{N}\sum_{i=1}^{N+1}\{\sigma(a(\boldsymbol{x}_{i,1} - \boldsymbol{x}_{*,1}) + 0.5) + \sigma(a(\boldsymbol{x}_{i,1} - \boldsymbol{x}_{*,1}) - 0.5)\}, \tag{24}$$

which results in

$$\tilde{\boldsymbol{h}}_j = \boldsymbol{h}_j + \frac{1}{N+1}\sum_{i=1}^{N+1}\sum_{m=\pm} \sigma(\langle \boldsymbol{Q}_m \boldsymbol{h}_i, \boldsymbol{K}_m \boldsymbol{h}_j \rangle)\boldsymbol{V}_m \boldsymbol{h}_j \tag{25}$$

$$= [\boldsymbol{x}_j, y_j, 1, t_j, u, \boldsymbol{0}, \boldsymbol{p}_j], \tag{26}$$

where $u = \hat{P}_z(t)(\boldsymbol{x}_{*,1})$, which can be regarded as a random variable sampled from $\mathcal{U}(0,1)$. $\quad\square$

*Proof of Theorem 5.* We divide a single update of Adam into the following three steps:

$$\boldsymbol{h}_n^{(2l)} = \begin{bmatrix} \boldsymbol{x}_i \\ y_i \\ 1 \\ u \\ \boldsymbol{b}_n \\ \hat{\boldsymbol{w}}^{(l)} \\ \boldsymbol{0} \\ \hat{\boldsymbol{m}}^{(l)} \\ \hat{\boldsymbol{v}}^{(l)} \\ \boldsymbol{p}_n \end{bmatrix} \xrightarrow{\text{Step 1}} \begin{bmatrix} \boldsymbol{x}_i \\ y_i \\ 1 \\ u \\ \boldsymbol{b}_n \\ \hat{\boldsymbol{w}}^{(l)} \\ \boldsymbol{g} \\ \beta_1 \hat{\boldsymbol{m}}^{(l)} \\ \beta_2 \hat{\boldsymbol{v}}^{(l)} \\ \boldsymbol{p}_n \end{bmatrix} \xrightarrow{\text{Step 2}} \begin{bmatrix} \boldsymbol{x}_i \\ y_i \\ 1 \\ u \\ \boldsymbol{b}_n \\ \hat{\boldsymbol{w}}^{(l)} \\ \boldsymbol{0} \\ \beta_1 \hat{\boldsymbol{m}}^{(l)} + (1-\beta_1)\boldsymbol{g} \\ \beta_2 \hat{\boldsymbol{v}}^{(l)} + (1-\beta_2)\boldsymbol{g}^2 \\ \boldsymbol{p}_n \end{bmatrix} \xrightarrow{\text{Step 3}} \begin{bmatrix} \boldsymbol{x}_i \\ y_i \\ 1 \\ u \\ \boldsymbol{b}_n \\ \hat{\boldsymbol{w}}^{(l)} - \eta\frac{\boldsymbol{m}^{(l+1)}/(1-\beta_1^l)}{\sqrt{\boldsymbol{v}^{(l+1)}/(1-\beta_2^l)} + \varepsilon\mathbf{1}} \\ \boldsymbol{0} \\ \hat{\boldsymbol{m}}^{(l+1)} \\ \hat{\boldsymbol{v}}^{(l+1)} \\ \boldsymbol{p}_n \end{bmatrix} = \tilde{\boldsymbol{h}}_i^{(2l+1)}, \tag{27}$$

where $\boldsymbol{g}$ indicates gradient. Step 1 is achieved in a single self-attention block, Step 2 is computed in a single feed-forward block, and finally, Step 3 is calculated in a feed-forward block. Thus, we need a two-layer Transformer for a single Adam step. Fix $\epsilon_1, \epsilon_2, \epsilon_3$ that are determined later.

**Step 1** As $\partial_1 \ell$ is $(\epsilon_1, R_1, M_1, C_1)$-approximable by sum of ReLUs, there exists a function $f : [-R_1, R_1]^2 \to \mathbb{R}$ of form

$$f(s,t) = \sum_{m=1}^{M_1} c_m \sigma(a_m s + b_m t + d_m), \tag{28}$$

with $\sum_{m=1}^{M_1} |c_m| \leq C, |a_m| + |b_m| + |d_m| \leq 1 (\forall m)$, such that $\sup_{(s,t) \in [-R_1, R_1]^2} |f(s,t) - \nabla_1 \ell(s,t)| \leq \epsilon_1$. Then, there exist matrices $\boldsymbol{Q}_m, \boldsymbol{K}_m, \boldsymbol{V}_m$ for $m \in \{1, \ldots, M_1\}$ such that

$$\boldsymbol{Q}_m \boldsymbol{h}_i = \begin{bmatrix} a_m \boldsymbol{w} \\ b_m \\ d_m \\ -2 \\ \boldsymbol{0} \end{bmatrix}, \quad \boldsymbol{K}_m \boldsymbol{h}_j = \begin{bmatrix} \boldsymbol{x}_j \\ y_j \\ 1 \\ R(1 - \boldsymbol{b}_{j,l}) \\ \boldsymbol{0} \end{bmatrix}, \text{ and } \boldsymbol{V}_m \boldsymbol{h}_j = \frac{(N+1)c_m}{N} \begin{bmatrix} \boldsymbol{0} \\ \boldsymbol{x}_j \\ \boldsymbol{0} \end{bmatrix}, \tag{29}$$

and $\boldsymbol{Q}_{M_1+1}, \boldsymbol{K}_{M_1+1}, \boldsymbol{V}_{M_1+1}$ such that

$$\boldsymbol{Q}_{M_1+1} \boldsymbol{h}_i = \begin{bmatrix} 1 \\ \boldsymbol{0} \end{bmatrix}, \boldsymbol{K}_{M_1+1} \boldsymbol{h}_j = \begin{bmatrix} 1 \\ \boldsymbol{0} \end{bmatrix}, \boldsymbol{V}_{M_1+1} \boldsymbol{h}_j = \begin{bmatrix} \boldsymbol{0} \\ \beta_1 \hat{\boldsymbol{m}}^{(l)} \\ \beta_2 \hat{\boldsymbol{v}}^{(l)} \\ \boldsymbol{0} \end{bmatrix}, \tag{30}$$

These matrices have norm bonds $\max_m \|\boldsymbol{Q}_m\| \leq 3, \max_m \|\boldsymbol{K}_m\| \leq 2 + R, \sum_m \|\boldsymbol{V}_m\| \leq 2C + (\beta_1 + \beta_2)$, for $m \in \{1, \ldots, M_1\}$. With these matrices, we get, for $m \in \{1, \ldots, M_1\}$,

$$\sigma(\langle \boldsymbol{Q}_m \boldsymbol{h}_i, \boldsymbol{K}_m \boldsymbol{h}_j \rangle) = \sigma(a_m \boldsymbol{w}^\top \boldsymbol{x}_j + b_m y_j + d_m) \mathbb{1}_{\boldsymbol{b}_{j,l}=1}, \tag{31}$$

and thus,

$$\frac{1}{N+1} \sum_{m=1}^{M_1+1} \sigma(\langle \boldsymbol{Q}_m \boldsymbol{h}_i, \boldsymbol{K}_m \boldsymbol{h}_j \rangle) \boldsymbol{V}_m \boldsymbol{h}_j \tag{32}$$

$$= \frac{1}{N} f(\boldsymbol{w}^\top \boldsymbol{x}_j, y_j) \mathbb{1}_{\boldsymbol{b}_{j,l}=1} [\boldsymbol{0}, \boldsymbol{x}_j, \boldsymbol{0}] + [\boldsymbol{0}, \beta_1 \hat{\boldsymbol{m}}^{(l)}, \beta_2 \hat{\boldsymbol{v}}^{(l)} \boldsymbol{0}] \tag{33}$$

$$= [\boldsymbol{0}, \boldsymbol{g}, \beta_1 \hat{\boldsymbol{m}}^{(l)}, \beta_2 \hat{\boldsymbol{v}}^{(l)}, \boldsymbol{0}]. \tag{34}$$

Finally, we get

$$\bar{\boldsymbol{h}}_n^{(2l)} := \text{Attn}(\boldsymbol{h}_n^{(2l)}) \tag{35}$$

$$= [\boldsymbol{x}_n, y_n, 1, u, \boldsymbol{b}_n, \hat{\boldsymbol{w}}^{(l)}, \boldsymbol{g}, \beta_1 \hat{\boldsymbol{m}}^{(l)}, \beta_2 \hat{\boldsymbol{v}}^{(l)}, \boldsymbol{p}_n]. \tag{36}$$

**Step 2.** As $s \mapsto s^2$ is $(\epsilon_2, R_2, M_2, C_2)$-approximable by sum of ReLUs, there exists a function $f : [-R_2, R_2] \to \mathbb{R}$ of form

$$f(s) = \sum_{m=1}^{M_2} c_m \sigma(a_m s + b_m), \tag{37}$$

with $\sum_{m=1}^{M_2} |c_m| \leq C, |a_m| + |b_m| \leq 1 (\forall m)$ such that $\sum_{s \in [-R_2, R_2]} |f(s) - s^2| \leq \epsilon_2$

With matrices $\boldsymbol{W}_1 \in \mathbb{R}^{3dM_2 \times D}$ and $\boldsymbol{W}_2 \in \mathbb{R}^{D \times 3dM_2}$, we get $\boldsymbol{W}_{1,m} \bar{\boldsymbol{h}}_n^{(2l)} = [a_m \boldsymbol{g} + b_m \mathbf{1}, \frac{1}{M_2} \boldsymbol{g}, -\frac{1}{M_2} \boldsymbol{g}]$ and $\boldsymbol{W}_2 \sigma(\boldsymbol{W}_1 \bar{\boldsymbol{h}}_n^{(2l)}) = [\boldsymbol{0}, -\boldsymbol{g}', (1-\beta_1)\boldsymbol{g}', (1-\beta_2) \sum_{m=1}^{M_2} c_m \sigma(a_m \boldsymbol{g} + b_m \mathbf{1}), \boldsymbol{0}]$, where $\boldsymbol{g}' = \sum_{m=1}^{M_2} \frac{1}{M_2} \{\sigma(\boldsymbol{g}) - \sigma(-\boldsymbol{g})\} = \boldsymbol{g}$. These matrices have norm bound of $\|\boldsymbol{W}_1\| + \|\boldsymbol{W}_2\| \leq 3 + \frac{2}{M_2} - \beta_1 + (1-\beta_2)C_2$. Consequently, we obtain

$$\tilde{\boldsymbol{h}}_n^{(2l)} = \text{MLP}(\bar{\boldsymbol{h}}_n^{(2l)}) \tag{38}$$

$$= [\boldsymbol{x}_n, y_n, 1, u, \boldsymbol{b}_n, \hat{\boldsymbol{w}}^{(l)}, \boldsymbol{g} - \boldsymbol{g}, \beta_1 \hat{\boldsymbol{m}}^{(l)} + (1-\beta_1)\boldsymbol{g}, \beta_2 \hat{\boldsymbol{v}}^{(l)} + (1-\beta_2)f(\boldsymbol{g}), \boldsymbol{p}_n], \tag{39}$$

where $\|f(\boldsymbol{g}) - \boldsymbol{g}^2\| \leq d\epsilon_3$. $\beta_1 \hat{\boldsymbol{m}}^{(l)} + (1-\beta_1)\boldsymbol{g}$ and $\beta_2 \hat{\boldsymbol{v}}^{(l)} + (1-\beta_2)\boldsymbol{g}^2$ are $\boldsymbol{m}^{(l+1)}$ and $\boldsymbol{v}^{(l+1)}$.

**Step 3.** As $(s,t) \mapsto \frac{s/(1-\beta_1^{(l)})}{\sqrt{t/(1-\beta_2^l)}+\varepsilon}$ is $(\epsilon_3, R_3, M_3, C_3)$-approximable by the sum of ReLUs, there exists a function $f$ as Equation (28) such that $\sum_{(s,t)\in[-R_3,R_3]^2}|f(s,t) - \frac{s/(1-\beta_1^{(l+1)})}{\sqrt{t/(1-\beta_2^l)}+\varepsilon}| \leq \epsilon_3$. With matrices $\boldsymbol{W}_1 \in \mathbb{R}^{dM_3 \times D}$ and $\boldsymbol{W}_2 \in \mathbb{R}^{D \times dM_3}$, we obtain

$$\boldsymbol{W}_{1,m}\bar{\boldsymbol{h}}_n^{(2l+1)} = [a_m \frac{\hat{\boldsymbol{m}}^{(l+1)}}{1-\beta_1^{(l+1)}} + b_m \frac{\hat{\boldsymbol{v}}^{(l+1)}}{1-\beta_2^{(l+1)}} + d_m \boldsymbol{1}] \tag{40}$$

and

$$\boldsymbol{W}_2\sigma(\boldsymbol{W}_1\bar{\boldsymbol{h}}_n^{(2l+1)}) = [\boldsymbol{0}, -\eta\sum_{m=1}^{M_3} c_m\sigma(a_m \frac{\hat{\boldsymbol{m}}^{(l+1)}}{1-\beta_1^{(l+1)}} + b_m \frac{\hat{\boldsymbol{v}}^{(l+1)}}{1-\beta_2^{(l+1)}} + d_m\boldsymbol{1}), \boldsymbol{0}]. \tag{41}$$

These matrices have norm bound of $\|\boldsymbol{W}_1\| + \|\boldsymbol{W}_2\| \leq \frac{1}{1-\max(\beta_1^{(l+1)},\beta_2^{(l+1)})} + \eta C_3$.

Finally, we get

$$\tilde{\boldsymbol{h}}_n^{(2l+1)} = \text{MLP}(\bar{\boldsymbol{h}}_n^{(2l+1)}) \tag{42}$$

$$= [\boldsymbol{x}_n, y_n, 1, u, \boldsymbol{b}_n, \hat{\boldsymbol{w}}^{(l)} - \boldsymbol{z}^{(l+1)}, \boldsymbol{0}, \hat{\boldsymbol{m}}^{(l+1)}, \hat{\boldsymbol{v}}^{(l+1)}, \boldsymbol{0}, \boldsymbol{p}_n], \tag{43}$$

where $\boldsymbol{z}^{(l)} = \eta f(\hat{\boldsymbol{m}}^{(l+1)}, \hat{\boldsymbol{v}}^{(l+1)})$ and $\|f(\hat{\boldsymbol{m}}^{(l+1)}, \hat{\boldsymbol{v}}^{(l+1)}) - \frac{\hat{\boldsymbol{m}}^{(l+1)}/(1-\beta_1^{(l)})}{\sqrt{\hat{\boldsymbol{v}}^{(l+1)}/(1-\beta_2^l)}+\varepsilon\boldsymbol{1}}\| \leq d\epsilon_3$.

To sum up, a single Adam step can be approximated with a two-layer Transformer with $M_1$ heads, $\max(3dM_2, dM_3)$ width MLP, and a norm of $\|\boldsymbol{\theta}\|_{\text{TF}} \leq \max\{5 + R + 2C + \beta_2 + \frac{2}{M_2} + (1-\beta_2)C_2, \frac{1}{1-\max(\beta_1,\beta_2)} + \eta C_3\}$. By appropriately selecting $\epsilon_1, \epsilon_2, \epsilon_3$, we have $\|\hat{\boldsymbol{w}}^{(l)} - \boldsymbol{w}_{\text{Adam}}^{(l)}\| \leq \epsilon l\eta B_x$. $\qquad\square$

