# OpenReview forum: "Transformers as Stochastic Optimizers"
_ICML.cc/2024/Workshop/ICL — ICML 2024 Workshop ICL Poster_

### Official Review · Reviewer_eTxN · 2024-06-09
**Review for the paper**

**Rating:** 2
**Fit:** 3
**Confidence:** 3

**Workshop Review:**

This paper makes a novel contribution by demonstrating that Transformers can act not only as deterministic, but also as stochastic optimizers within the in-context learning framework. The authors show that Transformers can generate both true and pseudo-random numbers, implement stochastic gradient descent (SGD), and even approximate more complex optimizers like Adam.

Specific comments:

1. l. 64: $\theta_1^{(l)}$... $\theta_M^{(l)}$ are not defined
2. Why encoding a position information is important. As far as I know, none of the previous works have done it.
3. Would it be possible to empirically find any of the constructions outlined in the paper? If so, this would straighten the contributions by a lot.

**Reason For Not Giving Higher Score:**

The paper presents a novel perspective on Transformers as stochastic in-context optimizers, potentially broadening the scope of in-context learning.

**Reason For Not Giving Lower Score:**

The paper lacks empirical validation to support its theoretical claims. The assumptions made, while reasonable, could be explored further in terms of practical limitations. The generalizability of the specific methods presented remains unclear.

---

### Official Review · Reviewer_kr9h · 2024-06-11
**Interesting contribution on pseudo-randomness for implementing stochastic gradient descent in-context**

**Rating:** 2
**Fit:** 3
**Confidence:** 2

**Workshop Review:**

The authors describe how Transformers can and may in practice implement stochastic gradient descent. The novelty here is a construction of how (pseudo-)randomness may be generated to support the stochastic component in the process. The papers does not seem to include the proofs for all relevant Theorems at this point (only Theorem 2), but has proof outlines. The constructions described and outlines in the paper seem credible so I would argue it makes a good poster discussion at the workshop. It certainly fits the topic of the workshop well.

Missing some important related works:

The mechanism of in-context learning (earlier referred to as memory-based meta-learning) has been described earlier in works such as [1]. Regarding neural networks that implement fairly general learning algorithms such as gradient descent in-context I recommend the authors also cite [2].

[1] Hochreiter, Sepp, A. Steven Younger, and Peter R. Conwell. "Learning to learn using gradient descent." Artificial Neural Networks—ICANN 2001: International Conference Vienna, Austria, August 21–25, 2001 Proceedings 11. Springer Berlin Heidelberg, 2001.
[2] Kirsch, Louis, and Jürgen Schmidhuber. "Meta learning backpropagation and improving it." Advances in Neural Information Processing Systems 34 (2021): 14122-14134.

**Reason For Not Giving Higher Score:**

The paper seems to be in a fairly early stage, this is a good fit for a workshop, but I would not recommend a spotlight talk at this point.

**Reason For Not Giving Lower Score:**

While some proofs are only sketched, the approach seems promising.

---

### Meta-Review · Area_Chair_vEfi · 2024-06-14

**Recommendation:** 2

**Metareview:**

The paper extends prior work demonstrating how transformers may implement gradient descent to the stochastic gradient descent setting. Notably, the authors demonstrate how random numbers may be generated based on the randomness in the data.

Both reviewers agree on the relevance and the quality of the submission. Reviewer kr9h points out that some missing related work and that the paper in its current state does not include all proofs for all theorems. Reviewer eTxN suggests to add empirical evidence to the claims made in the paper.

Overall, the paper is accepted as a poster.

---

### Decision · Program_Chairs · 2024-06-17

Accept (Poster)